# DIRseq as a method for predicting drug-interacting residues of intrinsically disordered proteins from sequences

**Matt MacAinsh[1], Sanbo Qin[1], Huan-Xiang Zhou[1,2]\***

[1]Department of Chemistry, University of Illinois Chicago, Chicago, United States;
[2]Department of Physics, University of Illinois Chicago, Chicago, United States

## eLife Assessment

This **important** study presents a sequence-based method for predicting drug-interacting residues in intrinsically disordered proteins (IDPs), addressing a significant challenge in understanding small-molecule:IDP interactions. The findings have **solid** support through examples underscoring the role of aromatic interactions. While predicted binding sites remain coarse, validation was done on a total of 10 IDPs at varying depths. The method builds on the authors' previous work and, with ad hoc modifications, is poised to benefit this emerging field.

**\*For correspondence:**
hzhou43@uic.edu

**Abstract** Intrinsically disordered proteins (IDPs) are now well-recognized as drug targets. Identifying drug-interacting residues is valuable for both optimizing compounds and elucidating the mechanism of action. Currently, NMR chemical shift perturbation and all-atom molecular dynamics (MD) simulations are the primary tools for this purpose. Here, we present DIRseq, a fast method for predicting drug-interacting residues from the amino-acid sequence. All residues contribute to the propensity of a particular residue to be drug-interacting; the contributing factor of each residue has an amplitude that is determined by its amino-acid type and attenuates with increasing sequence distance from the particular residue. DIRseq predictions match well with drug-interacting residues identified by NMR chemical shift perturbation and other methods, including residues $L_{22}WK_{24}$ and $Q_{52}WFT_{55}$ in the tumor suppressor protein p53. These successes augur well for deciphering the sequence code for IDP-drug binding. DIRseq is available as a web server at https://zhougroup-uic.github.io/DIRseq/ and has many applications, such as virtual screening against IDPs and designing IDP fragments for in-depth NMR and MD studies.

## Introduction

Intrinsically disordered proteins (IDPs) are now recognized as important drug targets (*Joshi and Vendruscolo, 2015*; *Saurabh et al., 2023*; *Wang et al., 2023*; *Uversky, 2024*). For structured protein targets, a crucial step in characterizing drug binding is identifying the drug-binding pocket. Although an IDP can be locked into a specific conformation by a drug molecule in rare cases (*Peterson et al., 2004*), the prevailing scenario is that the protein remains disordered upon drug binding (*Woods et al., 2011*; *Ono et al., 2012*; *Jin et al., 2013*; *De Mol et al., 2016*; *Heller et al., 2020*; *Iwaya et al., 2020*; *Zhao et al., 2021*; *Robustelli et al., 2022*; *Heller et al., 2024*). Consequently, the IDP-drug complex typically samples a vast conformational space, and the drug molecule only exhibits preferences, rather than exclusiveness, for interacting with subsets of residues (*Jin et al., 2013*; *Zhao et al., 2021*; *Robustelli et al., 2022*). Such drug-interacting residues, akin to binding pockets in structured proteins, are key to optimizing compounds (*Robustelli et al., 2022*; *Iconaru et al., 2015*; *Basu et al.,*

**Figure 1.** Four intrinsically disordered proteins (IDPs) and drugs that bind to them. (**a**) p27, p21, and SJ403. (**b**) p53 and epigallocatechin-3-gallate (EGCG). (**c**) α-Synuclein and Fasudil. For each IDP, the DIRseq propensities are rendered by a color spectrum from yellow for low values to red for high values. Predicted drug-interacting residues are shown with sidechains rendered in stick.

The online version of this article includes the following figure supplement(s) for figure 1:

**Figure supplement 1.** *q* parameters.

*2023*) and elucidating the mechanism of action (*Woods et al., 2011*; *De Mol et al., 2016*; *Zhao et al., 2021*; *Iconaru et al., 2015*; *Basu et al., 2023*).

NMR chemical shift perturbation (CSP) is the best experimental method for identifying drug-interacting residues and has been applied to many IDPs (*Ono et al., 2012*; *De Mol et al., 2016*; *Iwaya et al., 2020*; *Zhao et al., 2021*; *Robustelli et al., 2022*; *Heller et al., 2024*; *Iconaru et al., 2015*; *Basu et al., 2023*; *Hammoudeh et al., 2009*; *Lamberto et al., 2009*; *Krishnan et al., 2014*; *Tatenhorst et al., 2016*; *Ahmed et al., 2017*; *Kocis et al., 2017*; *Neira et al., 2017*; *Ruan et al., 2021*; *Iruela et al., 2022*; *Xu et al., 2022*). Aromatic residues are frequently among drug-interacting residues (*Zhao et al., 2021*; *Robustelli et al., 2022*; *Iconaru et al., 2015*; *Basu et al., 2023*; *Lamberto et al.,*

*2009*; *Tatenhorst et al., 2016*; *Ruan et al., 2021*). This is understandable as drug molecules typically are rich in aromatic rings (*Figure 1* and *Supplementary file 1A*), which can form π-π interactions with aromatic residues. As recently pointed out by *Heller et al., 2024*, CSPs of IDPs elicited by drug binding can be small enough to fall within the spectral resolution of NMR spectroscopy, therefore making it difficult to unequivocally identify drug-interacting residues. The small magnitude of CSPs may arise because the drug does not have a strong preference for interacting with any residues. Another scenario may be that drug binding induces a conformational shift such as secondary structure formation or even partial folding (*De Mol et al., 2016*; *Iwaya et al., 2020*; *Robustelli et al., 2022*), so CSPs spread from the directly interacting residues to the rest of the IDP, adding to the difficulty in identifying drug-interacting residues.

All-atom molecular dynamics (MD) simulations have presented atomic details in many IDP-drug systems (*Jin et al., 2013*; *Heller et al., 2020*; *Zhao et al., 2021*; *Robustelli et al., 2022*; *Basu et al., 2023*; *Ruan et al., 2021*; *Yu et al., 2016*; *Herrera-Nieto et al., 2020*; *Wang et al., 2022*; *Zhu et al., 2022*; *Mehta and Goyal, 2024*). These simulation studies have highlighted the frequent engagement of drug molecules with aromatic residues, particularly in π-π interactions (*Zhao et al., 2021*; *Robustelli et al., 2022*; *Basu et al., 2023*; *Ruan et al., 2021*; *Herrera-Nieto et al., 2020*; *Wang et al., 2022*; *Zhu et al., 2022*; *Mehta and Goyal, 2024*). MD simulations also revealed the emergence of a compact subpopulation of the N-terminal disordered region of the tumor suppressor protein p53 upon binding the drug epigallocatechin-3-gallate (EGCG) (*Zhao et al., 2021*) and reduced solvent exposure of hydrophobic residues in the 42-residue amyloid-β peptide (Aβ42) upon binding the drug 10074-G5 (*Heller et al., 2020*). However, MD simulations of IDPs still suffer from the perennial issues of inaccurate force fields and insufficient sampling, which are exacerbated when drug molecules are included. For example, two replicate simulations may identify different drug-interacting residues (*Lama et al., 2024*).

Here, we present a sequence-based method, DIRseq, as a complement to NMR CSP and MD simulations. This method was motivated by our observation that drug-interacting residues seem to overlap with residues exhibiting elevated transverse relaxation rates ($R_2$) (*Qin and Zhou, 2024*; *Muhammedkutty et al., 2025*), as exemplified by the C-terminal 20 residues of α-synuclein (*Robustelli et al., 2022*). Elevated $R_2$ in IDPs is caused by either local inter-residue interactions or residual secondary structures (*Dey et al., 2022*). Based on this understanding, we developed a sequence-based method, SeqDYN, to predict $R_2$ of IDPs (*Qin and Zhou, 2024*). As suggested previously (*Muhammedkutty et al., 2025*), the propensities of residues to form intramolecular interactions and therefore elevate $R_2$ should be similar to those for forming intermolecular interactions with drug molecules. DIRseq is therefore an adaptation of SeqDYN. DIRseq predictions match well with drug-interacting residues identified by NMR CSP and other methods, including residues $L_{22}WK_{24}$ and $Q_{52}WFT_{55}$ in p53 and C-terminal residues in α-synuclein. DIRseq is available as a web server at https://zhougroup-uic.github.io/DIRseq/ and has many applications.

## Results
### Retooling of SeqDYN into DIRseq

SeqDYN (*Qin and Zhou, 2024*) predicts the $R_2$ value of residue $n$ by accumulating a contributing factor $f(i; n)$ from each residue $i$:

$$R_2(n) = \Upsilon \prod_{i=1}^{N} f(i; n)$$

where $N$ is the total number of residues in the IDP and $\Upsilon$ is a uniform scale parameter. The contributing factor $f(i; n)$ has an amplitude $q(i)$ that is determined by the amino-acid type of residue $i$ and attenuates with increasing sequence distance, $s = |i - n|$, from residue $n$:

$$f(i; n) = 1 + \frac{q(i) - 1}{1 + bs^2}$$

The 20 $q$ parameters (one for each amino-acid type) and $b$ were trained on the measured $R_2$ values for 45 IDPs. The original $q$ values (*Figure 1—figure supplement 1a*) show that aromatic (Trp, Tyr, His, and Phe), Arg, and long aliphatic (Ile and Leu) amino acids are interaction-prone and tend to elevate

$R_2$. (The original SeqDYN also has an option of applying a helix boost; we did not apply this boost here.) We transformed SeqDYN into DIRseq by implementing four changes.

First, we reassigned four $q$ parameters. We lowered the $q$ values of the three long aliphatic amino acids, Leu, Ile, and Met, to the $q$ value of a short aliphatic amino acid, Val, because long aliphatic amino acids primarily participate in hydrophobic interactions, which may be less important for stabilizing the binding of a small molecule in sites largely exposed to water. At the same time, we increased the $q$ value of Asp to be the same as that of Glu, to increase the role of Asp's electrostatic interactions. Both the downgrade of hydrophobic interactions and the upgrade of electrostatic interactions were motivated by observations on drug binding to α-synuclein in MD simulations by *Robustelli et al., 2022*. The modified set of $q$ parameters is displayed in *Figure 1—figure supplement 1b*.

Second, the original $b$ value, 0.0316, corresponds to a correlation half-length, $b^{-1/2}$, of 5.6 residues. Given the small size of drug molecules, we increased the $b$ value to 0.3, corresponding to a correlation half-length of 1.8 residues. That is, we expect a drug molecule to interact with 3–4 residues at a time. Third, the SeqDYN-predicted $R_2$ profile, capturing experimental observations (*Klein-Seetharaman et al., 2002*), falls off at both termini, because no residues beyond the termini are present to provide a contributing factor to $R_2$. However, whereas intra-IDP interactions experience such a terminal effect, IDP-drug interactions do not. To eliminate the terminal effect, we padded the original IDP sequence by a stretch of 12 Gln residues at each terminus. Gln was selected because its $q$ value is at the middle of the 20 $q$ values (*Figure 1—figure supplement 1b*).

Lastly, we converted high and low $R_2$ values into high and low drug-interacting propensities, respectively, using a sigmoid function

$$P = \frac{100}{1 + e^{-\frac{R_2 - R_{2\text{th}}}{R_{2\text{wd}}}}}$$

where the midpoint $R_{2\text{th}}$ and width $R_{2\text{wd}}$ of the transition region are determined by the mean ($m$) and standard deviation (SD) of the $R_2$ values over the entire sequence. Specifically,

$$R_{2\text{th}} = m + s_1 \text{SD}$$

$$R_{2\text{wd}} = m/s_2$$

We chose $s_1$ to be 1.5 and $s_2$ to be 14.0. On the DIRseq web server, users can either keep these default values for $b$, $s_1$, and $s_2$, or enter values of their choice. The source code has been deposited on GitHub under the file name DIRseq.js (https://github.com/hzhou43/DIRseq/; *Zhou, 2025*). The values of $P$ range from 0 to 100; $P = 50$ when $R_2$ is at the 'threshold' value $R_{2\text{th}}$. Residues with $P \geq 50$ are predicted to be drug-interacting, but an isolated residue with $P \geq 50$ is discarded, as it can be reasonably expected that interactions with at least two consecutive (or nearby) residues are needed to generate sufficient drug binding stability.

## Detailed assessment of DIRseq on four IDPs

Drug-interacting residues in four IDPs or intrinsically disordered regions (IDRs) have been thoroughly characterized by NMR CSP. In *Figure 1* (left images), we show these IDPs in a single conformation, with DIRseq propensities rendered in a color spectrum (from low in yellow to high in red); predicted drug-interacting residues (i.e. those with $P \geq 50$) are shown with sidechains in stick. Drug molecules that bind to these IDPs are also displayed in *Figure 1* (right images). We display NMR CSPs as blue bars and DIRseq propensities as a red curve in *Figure 2*. Unless otherwise noted, we use $m+1.5$ SD as the threshold for identifying drug-interacting residues, for both CSP and DIRseq; this threshold is indicated by a horizontal dashed line in *Figure 2*. Experimentally identified drug-interacting residues are further indicated by cyan shading. Below, we present a detailed comparison between CSP and DIRseq.

The kinase inhibitory domain (residues 22–105) of the cell cycle regulator p27 harbors three aromatic-centered motifs that interact with a group of compounds represented by SJ403 (*Figure 1a*), as found by *Iconaru et al., 2015*. These motifs, $W_{60}N_{61}$, $E_{75}WQ_{77}$, and $Y_{88}Y_{89}$ (*Figure 2a*, cyan shading), are correctly predicted by DIRseq (*Figure 2a*, portions of the red curve above the 50% threshold). Specifically, the predicted drug-interacting residues are $R_{58}KWN_{61}$, $Y_{74}EWQ_{77}$, and $Y_{88}Y_{89}$ (*Figure 1a*,

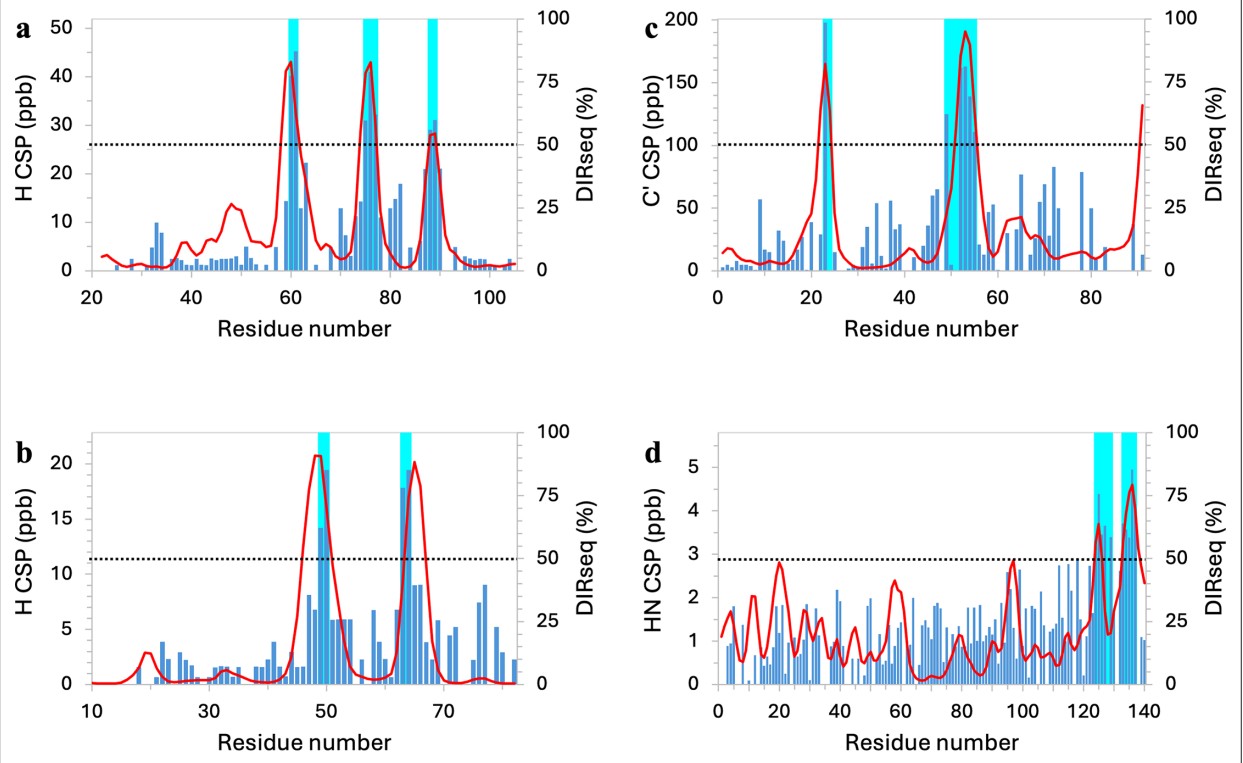

**Figure 2.** Comparison of DIRseq propensities with NMR chemical shift perturbations (CSPs). (**a**) p27. (**b**) p21. (**c**) p53. (**d**) α-Synuclein. CSPs are displayed as blue bars and in units of ppb; CSP-identified drug-interacting residues are indicated by cyan shading. DIRseq predictions are shown as red curves. The ordinate scales are chosen so that the $m+1.5$ SD threshold for CSP is at the same height as the 50% threshold for DIRseq, indicated by a horizontal dashed line.

The online version of this article includes the following source data and figure supplement(s) for figure 2:

**Source data 1.** Source data for *Figure 2*.

**Figure supplement 1.** Correlation between NMR chemical shift perturbation (CSP) and DIRseq propensity.

**Figure supplement 2.** Correlation of chemical shift perturbations (CSPs) elicited by two different compounds.

**Figure supplement 3.** Two compounds that bind to p27 with different characteristics.

left image). Moreover, the DIRseq propensity profile matches well the CSP profile along the sequence (*Figure 2a*). The two parameters show a very strong correlation, with a Pearson correlation coefficient (*r*) of 0.82 (*Figure 2—figure supplement 1a*).

Iconaru et al. also studied SJ403 binding to a related cell cycle regulator p21. Their CSP data identified the first two corresponding motifs, $W_{49}N_{50}$ and $F_{63}A_{64}$, but not the third as drug-interacting residues (*Figure 2b*, cyan shading). DIRseq correctly predicts both the first two motifs above the 50% threshold and the third motif below the threshold (*Figure 2b*, red curve; *Figure 1b*, left image). As noted by Iconaru et al., the p21 counterpart to the p27 third motif $F_{87}YY_{89}$ is $L_{76}YL_{78}$; the two Leu residues in p21 are no match for $F_{87}$ and $Y_{89}$ in p27. For the DIRseq method, Leu has a lower *q* value than Tyr and Phe (*Figure 1—figure supplement 1b*). CSP and DIRseq propensity show a strong correlation (*r*=0.66; *Figure 2—figure supplement 1b*).

NMR CSP revealed two motifs, $W_{23}K_{24}$ and $D_{49}IEQWFT_{55}$, in the N-terminal region of p53 as interacting with EGCG (*Zhao et al., 2021*; *Figure 2c*, cyan shading). MD simulations supported these drug-binding sites. DIRseq again correctly predicts these two motifs (*Figure 2c*, red curve). Specifically, the predicted drug-interacting motifs are $L_{22}WK_{24}$ and $E_{51}QWFT_{55}$ (*Figure 1c*, left image). Here also CSP and DIRseq propensity show a strong correlation (*r*=0.67; *Figure 2—figure supplement 1c*).

Using the $m+1.5$ SD threshold, CSPs of *Robustelli et al., 2022* select residues $A_{124}YEMPS_{129}$ and $Y_{133}QDYE_{137}$ of α-synuclein as interacting with the drug Fasudil (*Figure 2d*, cyan shading). These two C-terminal motifs are also identified by DIRseq, with residues $A_{124}YE_{126}$ and $Y_{133}QDYEP_{138}$ above the 50% threshold (*Figure 2d*, red curve; *Figure 1d*, left image). There is a moderate correlation (*r*=0.51;

*Figure 2—figure supplement 1d*) between CSP and DIRseq propensity. Robustelli et al. further characterized the IDP-drug interactions by all-atom simulations, using both full-length α-synuclein and a C-terminal fragment (residues 121–140). Further simulations of the C-terminal fragment binding with additional compounds led to a compound known as Ligand 47 with a somewhat higher affinity than Fasudil. The CSP profiles elicited by Fasudil and Ligand 47 are similar, though with larger amplitudes around $Y_{125}$ and $Y_{136}$ by the latter; their correlation coefficient is 0.78 (*Figure 2—figure supplement 2a*). Because DIRseq does not consider any information about the drug molecule, this correlation coefficient between CSPs elicited by two different compounds might be viewed as an upper bound of what can be achieved for the correlation between CSP and DIRseq propensity.

## Dependences of DIRseq prediction accuracies on model parameters

We now use the CSP data of the above four IDPs to assess the dependences of DIRseq prediction accuracies on model parameters (*Supplementary file 1B*). We use two complementary measures for accuracy: the Pearson correlation between CSP and drug-interacting propensity and the difference between true positive (TP) and false positive (FP). The four IDPs have a total of 31 CSP-identified drug-interacting residues. Recall that DIRseq uses a sigmoid function to convert the $R_2$ output of SeqDYN into drug-interacting propensities. Using the original 20 $q$ parameters along with the present values for $b$, $s_1$, and $s_2$, the predictions are already pretty reasonable: the correlations for the four IDPs range from 0.46 to 0.79 (sum = 2.43) and TP outnumbers FP by 7 residues. Indeed, this initial success was anticipated (*Qin and Zhou, 2024*; *Muhammedkutty et al., 2025*) and validated the premise of DIRseq, i.e., intra-IDP interactions that elevate $R_2$ also tend to mediate drug interactions. As stated above, we tweaked 4 of the 20 $q$ parameters to arrive at the final DIRseq; now the $r$ sum increases to 2.66 and TP – FP increases to 15.

Examining the four $q$ parameter changes one at a time, the downgrade of $q$ for Leu increases both $r$ sum (to 2.53) and TP – FP (to 16) relative to the counterparts with the original SeqDYN $q$ parameters. In comparison, the downgrade of $q$ for Ile does not affect $r$ sum and yields a small increase in TP – FP (to 10), the upgrade of $q$ for Asp yields small increases in both $r$ sum (to 2.52) and TP – FP (to 8), but the downgrade of $q$ for Met actually decreases both $r$ sum (to 2.37) and TP – FP (to 6). The latter result shows that we made these parameter tweaks not solely for increasing accuracy but for physical reasons, i.e., to reduce the role of hydrophobic interactions but elevate the role of electrostatic interactions in IDP-drug binding, as suggested by MD simulations.

We also tested alternative models for $q$ parameters. Given the prominence of aromatic amino acids in drug binding as revealed by CSP, we wondered how a model that solely emphasizes aromatic amino acids would perform. To that end, we tested a model with only two different $q$ values for the 20 amino acids: a high value (same as that of Trp in SeqDYN) for all three aromatic amino acids (Trp, Tyr, and Phe) and a low value (same as that of Gly in SeqDYN) for all other amino acids. This is similar to a sticker-spacer model for simulating liquid-liquid phase separation (*Martin et al., 2020*). This aromatic model achieves the same $r$ sum, 2.66, as DIRseq, but its FP outnumbers its TP, such that TP – FP = –1. *Tesei and Lindorff-Larsen, 2022* parameterized a coarse-grained force field to simulate liquid-liquid phase separation. Their λ 'stickiness' parameters have a good correlation with the SeqDYN original $q$ parameters (*Zhou et al., 2024*). We replaced the $q$ parameters by these λ parameters (after scaling to the same numerical range as the $q$ parameters); the resulting model has both a low $r$ sum (1.21) and a low TP – FP (–21). This outcome suggests that, unlike liquid-liquid phase separation where stickiness is the main drive, drug binding is more selective in the type of intermolecular interactions. Lastly, we tested a model where an average hydropathy scale (compiled by *Tesei et al., 2021*) was used in place of the $q$ parameters; the resulting model has very little predictive value ($r$ sum = 0.44 and TP – FP = –20). This last outcome is in line with our downgrade of hydrophobic interactions in DIRseq. The test results from all these alternative models indicate that the $q$ parameters in the final DIRseq model capture the appropriate balance among aromatic, electrostatic, and hydrophobic interactions in IDP-drug binding.

In addition to the 20 $q$ parameters, DIRseq has 3 other parameters. The $b$ parameter determines the number of residues that simultaneously interact with a drug molecule. In *Supplementary file 1*, we list the performance measures when $b$ is varied. These results show that our final choice, 0.3, for $b$ is optimal for both $r$ sum and TP – FP. The corresponding number of residues, 3–4, that simultaneously interact with a drug molecule fits with the narrow range of drug molecule sizes in the present

study (*Supplementary file 1A*; molecular weights: 360±130 Da). The last two parameters, $s_1$ and $s_2$, are in the sigmoid function that converts the $R_2$ output into drug-interacting propensities. $s_1$ sets the $R_2$ threshold for labeling a residue as drug-interacting. Again, our final choice, corresponding to $m$+1.5 SD for the threshold, achieves an optimum for $r$ sum and TP − FP. A lower threshold leads to a high FP and also a slight deterioration in $r$ sum, whereas a higher threshold leads to a low FP and possibly a minuscule improvement in $r$ sum. $s_2$ controls the sharpness of the sigmoid function in the transition region and affects $r$ sum but not TP − FP. $r$ sum increases with increasing $s_2$; we chose an $s_2$ value, where $r$ sum is nearly at the saturation level.

## DIRseq as a complement to CSP for assigning drug-interacting residues

After assessing the achievable accuracy of DIRseq, we now consider an application: its combination with CSP to make robust assignments of drug-interacting residues. We note that the above CSP-identified drug-interacting motifs are all anchored on one or more aromatic residues, and this feature likely contributes to the good performance of DIRseq. When a clear 'aromatic signal' is not present, CSP or DIRseq alone may not be able to conclusively identify drug-interacting residues. However, a consensus identification by CSP and DIRseq may be reliable. MD simulations have played such a complementary role to CSP in several studies (*Zhao et al., 2021*; *Robustelli et al., 2022*; *Basu et al., 2023*; *Zhu et al., 2022*).

*De Mol et al., 2016* reported the CSPs of an IDR called AF-1* (residues 142–448) in the androgen receptor elicited by EPI-001 and its stereoisomers, including EPI-002; small but reproducible CSPs were found in three subregions, R1 (residues 341–371), R2 (residues 391–414), and R3 (residues 426–446). Conversely, AF-1* caused changes in the $^1$H NMR spectrum of EPI-001, but the individual peptides corresponding to R1, R2, and R3 did not, suggesting that, rather than separately interacting with the three subregions, EPI-001 induces a partial folding of the three subregions. MD simulations captured the partial folding of the R2-R3 fragment (residues 391–446) induced by EPI-002 and under-scored the importance of aromatic residues in drug binding (*Zhu et al., 2022*). *Basu et al., 2023* reported the CSPs of the transactivation unit 5 (Tau-5*, residues 336–448) in AF-1* by EPI-001 and a more potent variant, 1aa (*Figure 3a*, *Figure 2—figure supplement 2b*). The contact probabilities of the R2-R3 fragment with 1aa in MD simulations reaffirmed the importance of aromatic residues.

*De Mol et al., 2016* and *Basu et al., 2023* were careful not to name any drug-interacting residues based on CSPs. In *Figure 3a*, we compare the 1aa-elicited CSPs and DIRseq propensities of Tau-5*. In agreement with both NMR studies, DIRseq identifies drug-interacting residues in the middle of each of R1-R3: $R_{360}DYY_{363}$, $A_{396}WAA_{399}$, and $W_{433}H_{434}$. In addition, the latter two motifs showed the highest drug-contact probabilities in separate MD simulations of the R2-R3 fragment (*Basu et al., 2023*; *Zhu et al., 2022*). For the R2 and R3 subregions, CSPs above the $m$+1.5 SD threshold are observed at residues downstream of the DIRseq identifications, so we propose to expand the drug-interacting motifs to $A_{396}WAAAAAQ_{403}$ and $W_{433}HTLF_{437}$. The three putative drug-interacting motifs are indicated as cyan shading in *Figure 3a*.

*Heller et al., 2024* used $^{19}$F transverse relaxation measurements to determine the binding affinity of the disordered domains 2 and 3 of the hepatitis C virus NS5A protein (NS5A-D2D3, residues 247–466) for 5-fluorindole. They also measured $^1$H-$^{15}$N CSPs at two ligand concentrations but described them as 'nearly undetectable' (*Figure 3b*). We speculate that the small CSPs may be due to the small size of the ligand, making it difficult to interact with multiple residues simultaneously and thus achieve sufficient binding stability. In any event, CSPs above the $m$+1.5 SD threshold were largely isolated (and thus appeared to be random), and there was not much overlap between residues having these above-the-threshold CSPs at the two ligand concentrations. The one exception is a motif around $W_{312}$, which had above-the-threshold CSPs at both ligand concentrations, and nearby residues $A_{308}$ and $A_{313}$ also had above-the-threshold CSPs at one of the ligand concentrations. DIRseq predicts the motif $P_{310}AWARPD_{316}$ as drug-interacting residues. We propose the expanded motif, $A_{308}LPAWARPD_{316}$, as residues interacting with 5-fluorindole (*Figure 3b*, cyan shading). DIRseq also predicts two more motifs, $E_{323}SWRRPDY_{330}$ and $R_{352}RRR_{355}$, as drug-interacting residues, which remain to be tested.

The fibrillation of acid-denatured $\beta_2$ microglobulin is inhibited by rifamycin SV (*Woods et al., 2011*). An aromatic-rich motif, $W_{60}SFYLLYYTEF_{70}$, was implicated in the nucleation of fibrillation and also involved in ligand binding, as a triple mutation, F62A/Y63A/Y67A, significantly weakened binding. Low intensities of NMR peaks from residues 58–79 (possibly due to the formation of residual

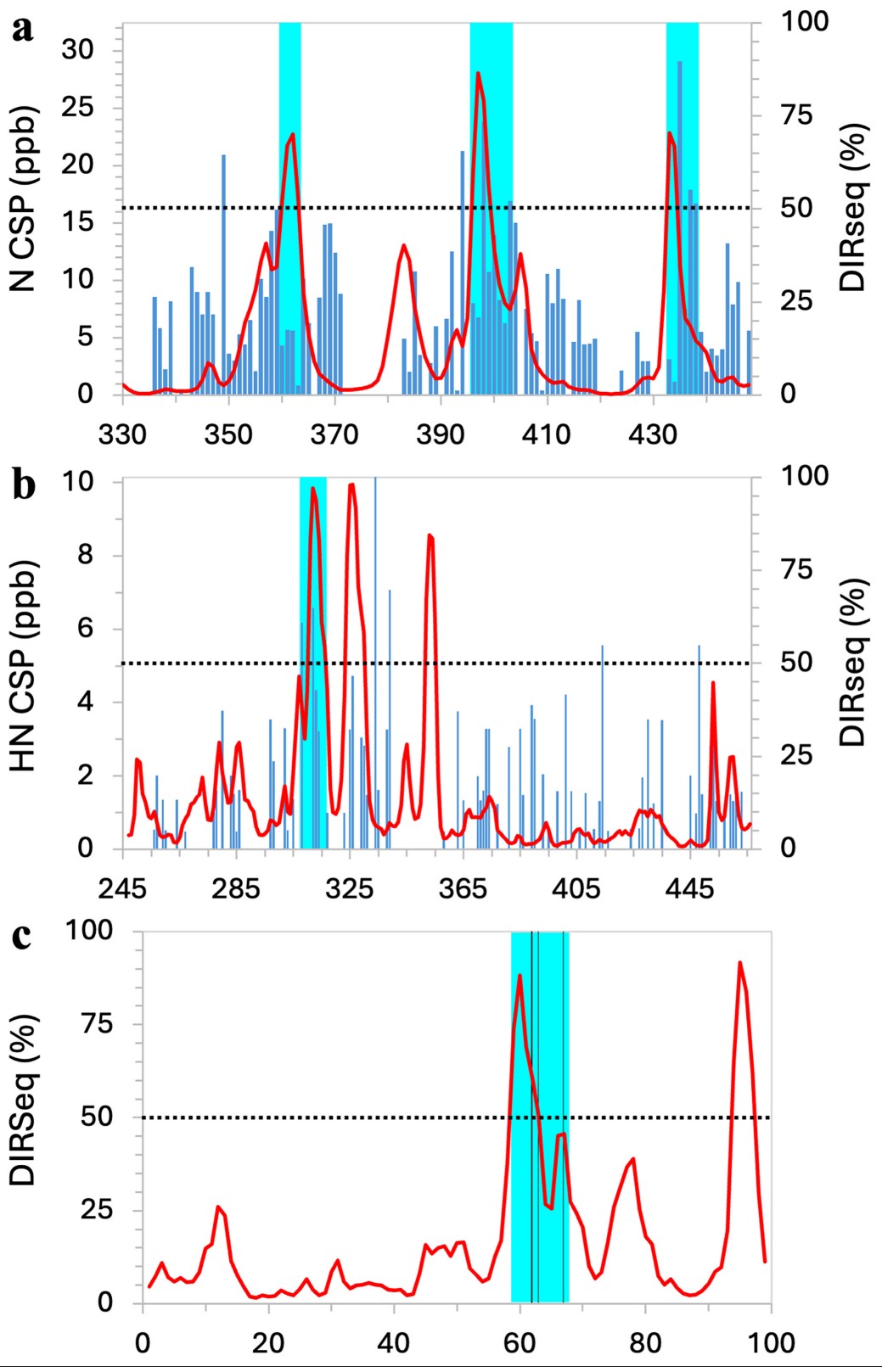

**Figure 3.** Drug-binding sites identified by combining chemical shift perturbation (CSP) or mutation data with DIRseq predictions. (**a**) Tau-5*. (**b**) NS5A-D2D3. (**c**) β2 microglobulin. Display items in panels (**a, b**) have the same meanings as in *Figure 2*, except that cyan shading indicates consensus identification; in panel (**c**), vertical lines indicate mutation sites.

*Figure 3 continued on next page*

*Figure 3 continued*

The online version of this article includes the following source data for figure 3:

**Source data 1.** Source data for *Figure 3*.

structures) prevented the measurement of CSPs. DIRseq predicts $D_{59}WSFY_{63}$ as drug-interacting residues. Combined with the mutational data of *Woods et al., 2011*, we propose the expanded motif, $D_{59}WSFYLLYY_{67}$, as the major drug-binding site (*Figure 3c*, cyan shading). DIRseq also predicts an additional motif, $K_{94}WDR_{97}$, as drug-interacting residues.

The aggregation of human islet amyloid polypeptide (hIAPP; 37 residues) is inhibited by the small molecule YX-I-1 (*Xu et al., 2022*). CSPs elicited by this molecule were small (*Figure 4a*). In addition, CSPs of short IDPs may not exhibit strong disparities because amino acids may be too well mixed along the sequence or drug binding may induce a conformational shift. We thus reduce the threshold for identifying drug-interacting residues to $m+1.0$ SD when the number of residues is $\leq 50$. With this threshold, three residues are identified by CSP as drug-interacting residues: $R_{11}$ and $V_{17}H_{18}$. In comparison, DIRseq identifies $T_9QRLA_{13}$ and $F_{15}$ as drug-interacting residues, which partially overlap with the CSP identifications. Combining the two types of data, we propose $T_9QRLANFLVH_{18}$ as the primary drug-binding site (*Figure 4a*, cyan shading). We note that this motif is also prone to $\alpha$-helix formation (*Apostolidou et al., 2008*).

*Ono et al., 2012* acquired $^1H$-$^{15}N$ heteronuclear single quantum coherence spectra of the 42-residue amyloid-$\beta$ (A$\beta$42) in the absence and presence of the oligomerization-blocking compound myricetin. CSPs were not calculated, but chemical shift movements were most pronounced at $R_5$, $V_{12}H_{13}$, $K_{16}LVF$-$FAED_{23}$, and $I_{31}I_{32}$ (vertical lines in *Figure 4b*). In addition, NMR cross peaks suffered broadening upon ligand binding at four of these residues: $R_5$, $V_{12}$, $K_{16}$, and $V_{18}$ (dark vertical lines in *Figure 4b*), implicating elevated probabilities of ligand interactions. DIRseq predicts $E_3FRH_6$ and $H_{13}H_{14}$ as drug-interacting residues. Combined with the NMR data, we propose $E_3FRH_6$ and $V_{12}HHQKLV_{18}$ as the primary ligand-binding sites (*Figure 4b*, cyan shading). A$\beta$42 CSPs elicited by several other compounds were also widely distributed over the sequence, such that *Iwaya et al., 2020* 'failed to identify' drug-interacting residues, implicating a conformational shift.

*Hammoudeh et al., 2009* identified three distinct drug binding sites in a C-terminal IDR (residues 363–412) of the oncoprotein c-Myc. This region is disordered on its own but forms a helix-loop-helix structure upon heterodimerization with Max. These authors measured the CSPs of three overlapping fragments (residues 363–381, 370–409, and 402–412), each in the presence of a single compound and of the full IDR in the presence of all three compounds. Using 20 parts per billion (ppb) as the threshold, Hammoudeh et al. named residues $R_{366}RNELKRSFF_{375}$, $F_{375}ALRDQIPELE_{385}$, and $Y_{402}ILSVQAE_{409}$ as the binding sites for 10074-G5, 10074-A4, and 10058-F4, respectively. We present their CSP data for the full IDR in *Figure 4c*. Using the $m+1.0$ SD threshold, only six residues are identified as drug-interacting residues: $R_{367}$, $L_{370}$, $F_{374}F_{375}$, $A_{399}$, and $S_{405}$. DIRseq predicts two motifs, $E_{363}RQRR_{367}$ and $K_{371}RSFFA_{376}$, that overlap with the first two sites identified by CSP; a third motif, $K_{397}KATAY_{402}$, that corresponds to the third CSP-identified site has moderate drug-interacting propensities. Combining CSP and DIRseq, we revise the three drug-binding sites to be $E_{363}RQRRNELKRSFF_{375}$, $S_{373}FFALRDQI_{381}$, and $K_{397}KATAYILS_{405}$ (*Figure 4c*, cyan, olive, and yellow shading, respectively). In addition to 10074-G5, 10074-A4, and 10058-F4, many compounds bind to these three sites (*Yu et al., 2016*; *Han et al., 2019*; *Shirey et al., 2021*; *Li et al., 2024*).

## Discussion

We have presented the first sequence-based method, DIRseq, to predict drug-interacting residues of IDPs. Assessment against NMR CSP demonstrates the accuracy of DIRseq. Drug-binding motifs are anchored on one or more aromatic residues for forming $\pi$-$\pi$ interactions with drug-like molecules that are rich in aromatic groups. The success of DIRseq comes without any specific information on the drug molecules, suggesting that IDPs may have a relatively simple sequence code for drug binding.

The notion that drug-interacting residues may be agnostic to the molecular details of drug compounds is supported by the fact that the same drug can bind to different IDPs. For example, in addition to p53 (*Zhao et al., 2021*; *Figure 2c*), the polyphenol EGCG also binds to many other IDPs, including $\alpha$-synuclein (*Ehrnhoefer et al., 2008*), hIAPP (*Young et al., 2014*; *Meng et al., 2010*), A$\beta$40

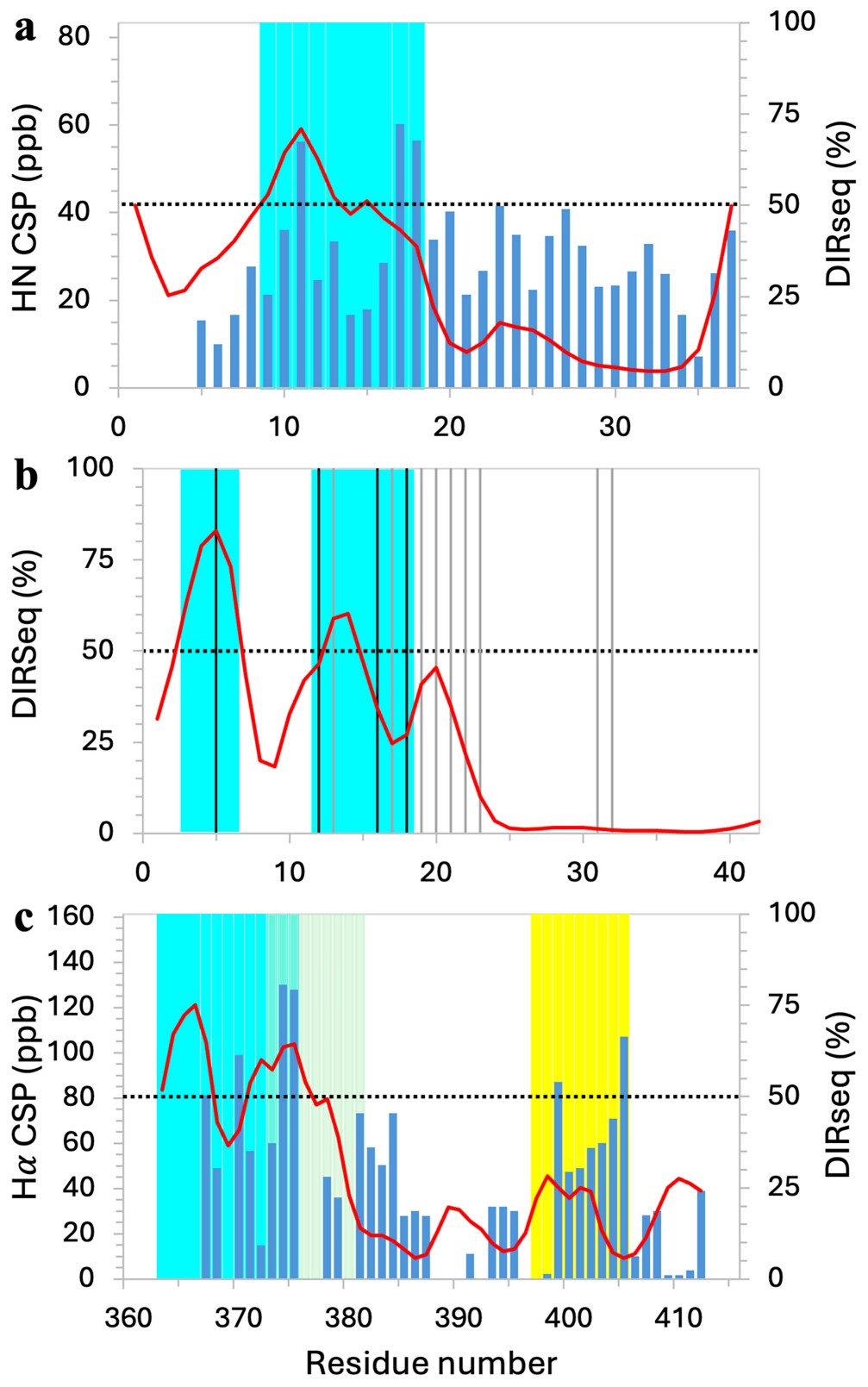

**Figure 4.** Drug-binding sites identified by combining chemical shift perturbation (CSP) or mutation data with DIRseq predictions. (**a**) hIAPP. (**b**) Aβ42. (**c**) c-Myc. Display items have the same meanings as in *Figure 2*, but with the following exceptions. (1) In panel (**b**), vertical lines indicate residues with prominent CSPs; those accompanied by NMR peak broadening have their vertical lines in dark color. (2) In panel (**c**), three CSP-identified

*Figure 4 continued on next page*

*Figure 4 continued*

drug-interacting regions are indicated by cyan, olive, and yellow shading. (3) The threshold for identifying drug-interacting residues is lowered to $m+1.0$ SD.

The online version of this article includes the following source data for figure 4:

**Source data 1.** Source data for *Figure 4*.

---

(*Ahmed et al., 2017*) and Aβ42 (*Ehrnhoefer et al., 2008*), tau (*Wobst et al., 2015*), and merozoite surface protein 2 (*Chandrashekaran et al., 2010*). Likewise, 10074-G5 binds to c-Myc (*Hammoudeh et al., 2009*; *Figure 4c*) but also to Aβ42 (*Heller et al., 2020*). On the other hand, c-Myc represents a case where different compounds bind to distinct sites on a single IDP (*Hammoudeh et al., 2009*). A related example is presented by p27, where SJ403 typifies a group of compounds that share the same three binding sites (*Figure 2a*). Another group of compounds, typified by SJ710, binds only to the third site. Chemically, the presence of nitrogen atoms in the rings of SJ403 enhances its aromaticity and thus strengthens π-π interactions; in addition, the electronegative groups of SJ403 project into different directions, making it less restricted when forming electrostatic interactions (*Figure 2—figure supplement 3*). These features may explain why SJ403 can bind to all three sites, whereas SJ710 can bind only to the third site, $F_{87}YYR_{90}$, where three consecutive aromatic residues followed by a basic residue ensure that SJ710 can form both π-π and electrostatic interactions. When more data for multiple drugs binding to a single IDP becomes available, it will be important to use such data to train the next generation of DIRseq where the parameters are drug-specific. As a simple example, the number of residues that can simultaneously bind a drug molecule may grow with the latter's size; this dependence can be modeled by making the parameter $b$ dependent on drug molecule size. The drug molecules studied in the present work have molecular weights of 360±130 Da and thus span a relatively narrow size range.

We have illustrated the combination of DIRseq with NMR CSP to make robust identifications of drug-binding sites in IDPs. Indeed, CSP, MD simulations, and DIRseq are three orthogonal approaches that have great potential in complementing each other, not only for identifying drug-binding sites but also for elucidating the roles of amino acids, their sequence context, and different types of noncovalent

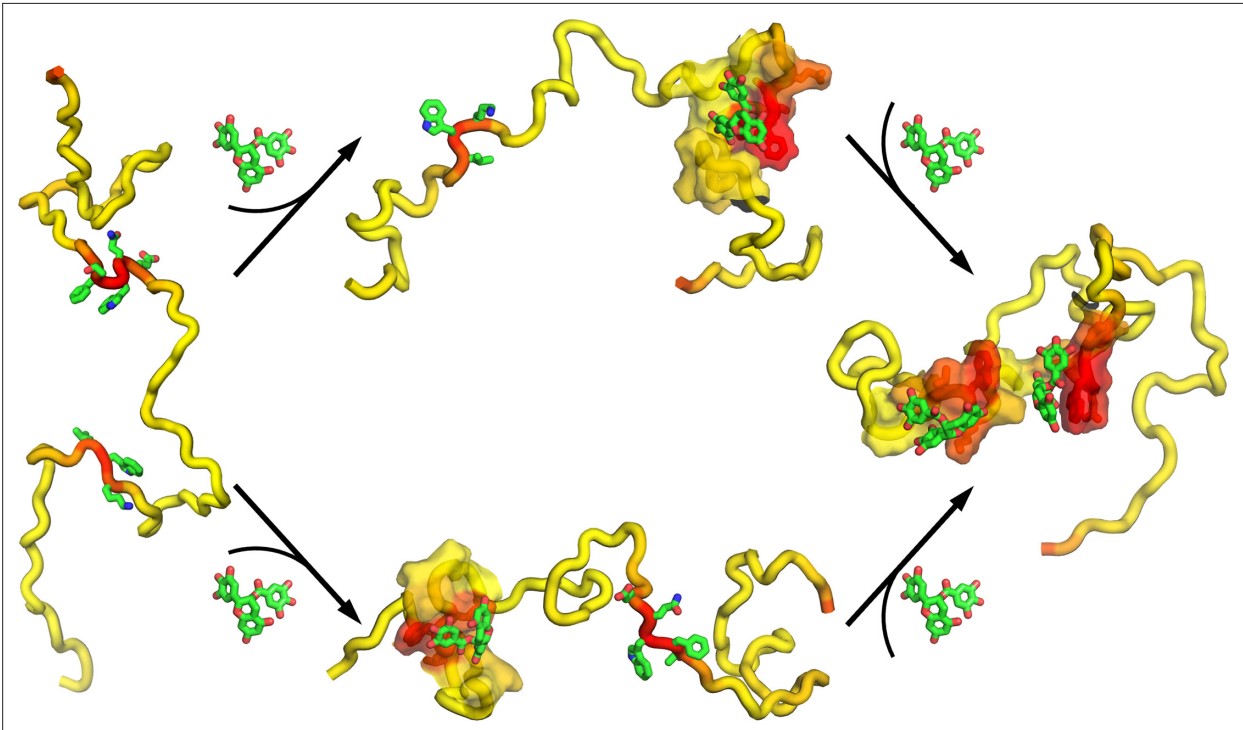

**Figure 5.** Poses of p53-bound EGCG generated by docking.

interactions in forming such sites. DIRseq offers fast speed and a simple, direct link between sequence motif and propensity for drug binding.

Another application of DIRseq is to define IDP fragments for in-depth study by MD simulations, as shorter constructs both enable the use of a smaller simulation box and reduce the size of the conformational space. For example, based on CSP data and initial MD simulations of full-length α-synuclein, *Robustelli et al., 2022* chose a 20-residue C-terminal fragment for simulations of binding with additional compounds, leading to the identification of Ligand 47 as a stronger binder than the original Fasudil. Similarly, based on CSP data from longer constructs of the androgen receptor, both *Zhu et al., 2022* and *Basu et al., 2023* chose the 56-residue R2-R3 fragment for MD simulations of drug binding. DIRseq can now play a similar role in selecting fragments for MD simulations when CSP data are unavailable. Longer constructs may also present challenges such as resonance assignments to NMR experiments, so well-chosen fragments guided by DIRseq can also benefit NMR studies.

Lastly, virtual screening has been conducted against conformational ensembles of IDPs (*Ruan et al., 2021*; *Dhar et al., 2025*); drug-binding sites predicted by DIRseq can be used to guide such screening. As a simple illustration, we present poses of EGCG generated by screening against the two DIRseq-predicted binding sites in p53 in *Figure 5*. As IDPs sample a vast conformational space, knowledge of the binding site can drastically reduce the computational cost. The subset of conformations that generate high docking scores for a given drug at the known site can also provide insight into the mechanism of drug action.

## Methods

The sequences of the IDP studied here and the drugs that bind to them are listed in *Supplementary file 1A*. All DIRseq predictions were obtained using the web server at https://zhougroup-uic.github.io/DIRseq/. Conformations of IDPs were generated using the TraDES method (*Feldman and Hogue, 2002*).

Docking of EGCG onto p53 was performed via the SwissDock web server at https://www.swissdock.ch/ (*Bugnon et al., 2024*) utilizing the Autodock Vina docking engine (*Eberhardt et al., 2021*). The SMILES string for EGCG from PubChem (CID 65064) and several conformations of p53 were used as input. A cubic region (13–20 Å in side length) around the center of each drug-interacting residue was selected for docking.

## Acknowledgements

This work was supported by National Institutes of Health Grant GM118091.

## Additional information

### Funding

| Funder | Grant reference number | Author |
|---|---|---|
| National Institute of General Medical Sciences | GM118091 | Huan-Xiang Zhou |

The funders had no role in study design, data collection and interpretation, or the decision to submit the work for publication.

### Author contributions
Matt MacAinsh, Data curation, Formal analysis, Validation, Investigation, Methodology; Sanbo Qin, Software, Visualization; Huan-Xiang Zhou, Conceptualization, Data curation, Formal analysis, Supervision, Funding acquisition, Investigation, Visualization, Methodology, Writing – original draft, Project administration, Writing – review and editing

### Author ORCIDs
Huan-Xiang Zhou ⬥ https://orcid.org/0000-0001-9020-0302

Reviewer #1 (Public review): https://doi.org/10.7554/eLife.107470.3.sa1
Reviewer #2 (Public review): https://doi.org/10.7554/eLife.107470.3.sa2
Author response https://doi.org/10.7554/eLife.107470.3.sa3

## Additional files

### Supplementary files
Supplementary file 1. Supplementary tables.

MDAR checklist

### Data availability
Figure 2-source data 1, Figure 3-source data 1, Figure 4-source data 1 contain the numerical data used to generate the figures. The source code for DIRseq can be downloaded at https://github.com/hzhou43/DIRseq/ with file name DIRseq.js (copy archived at *Zhou, 2025*).

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
