## [Editor Report · eLife Assessment]

This **important** study presents a sequence-based method for predicting drug-interacting residues in intrinsically disordered proteins (IDPs), addressing a significant challenge in understanding small-molecule:IDP interactions. The findings have **solid** support through examples underscoring the role of aromatic interactions. While predicted binding sites remain coarse, validation was done on a total of 10 IDPs at varying depths. The method builds on the authors' previous work and, with ad hoc modifications, is poised to benefit this emerging field.

---

## [Referee Report · Reviewer #1 (Public review)]

Summary:

The authors developed a sequence-based method to predict drug-interacting residues in IDP, based on their recent work, to predict the transverse relaxation rates (R2) of IDP trained on 45 IDP sequences and their corresponding R2 values. The discovery is that the IDPs interact with drugs mostly using aromatic residues that are easy to understand, as most drugs contain aromatic rings. They validated the method using several case studies, and the predictions are in accordance with chemical shift perturbations and MD simulations. The location of the predicted residues serves as a starting point for ligand optimization.

Strengths:

This work provides the first sequence-based prediction method to identify potential drug-interacting residues in IDP. The validity of the method is supported by case studies. It is easy to use, and no time-consuming MD simulations and NMR studies are needed.

Weaknesses:

The method does not depend on the information of binding compounds, which may give general features of IDP-drug binding. However, due to the size and chemical structures of the compounds (for example, how many aromatic rings), the number of interacting residues varies, which is not considered in this work. Lacking specific information may restrict its application in compound optimization, aiming to derive specific and potent binding compounds.

Comments on revised version:

I'm satisfied with the authors' response and the public review does not need further changes.

---

## [Referee Report · Reviewer #2 (Public review)]

Summary:

In this work, the authors introduce DIRseq, a fast, sequence-based method that predicts drug-interacting residues (DIRs) in IDPs without requiring structural or drug information. DIRseq builds on the authors' prior work looking at NMR relaxation rates, and presumes that those residues that show enhanced R2 values are the residues that will interact with drugs, allowing these residues to be nominated from the sequence directly. By making small modifications to their prior tool, DIRseq enables the prediction of residues seen to interact with small molecules in vivo.

Strengths:

The preprint is well written and easy to follow.

---

## [Author Response]

The following is the authors’ response to the original reviews.

**Reviewer #1 (Public review):**
Summary:The authors developed a sequence-based method to predict drug-interacting residues in IDP, based on their recent work, to predict the transverse relaxation rates (R2) of IDP trained on 45 IDP sequences and their corresponding R2 values. The discovery is that the IDPs interact with drugs mostly using aromatic residues that are easy to understand, as most drugs contain aromatic rings. They validated the method using several case studies, and the predictions are in accordance with chemical shift perturbations and MD simulations. The location of the predicted residues serves as a starting point for ligand optimization.Strengths:This work provides the first sequence-based prediction method to identify potential druginteracting residues in IDP. The validity of the method is supported by case studies. It is easy to use, and no time-consuming MD simulations and NMR studies are needed.Weaknesses:The method does not depend on the information of binding compounds, which may give general features of IDP-drug binding. However, due to the size and chemical structures of the compounds (for example, how many aromatic rings), the number of interacting residues varies, which is not considered in this work. Lacking specific information may restrict its application in compound optimization, aiming to derive specific and potent binding compounds.

We fully recognize that different compounds may have different interaction propensity profiles along the IDP sequence. In future studies, we will investigate compound-specific parameter values. The limiting factor is training data, but such data are beginning to be available.

**Reviewer #2 (Public review):**
Summary:In this work, the authors introduce DIRseq, a fast, sequence-based method that predicts druginteracting residues (DIRs) in IDPs without requiring structural or drug information. DIRseq builds on the authors' prior work looking at NMR relaxation rates, and presumes that those residues that show enhanced R2 values are the residues that will interact with drugs, allowing these residues to be nominated from the sequence directly. By making small modifications to their prior tool, DIRseq enables the prediction of residues seen to interact with small molecules in vivo.Strengths:The preprint is well written and easy to followWeaknesses:(1) The DIRseq method is based on SeqDYN, which itself is a simple (which I do not mean as a negative - simple is good!) statistical predictor for R2 relaxation rates. The challenge here is that R2 rates cover a range of timescales, so the physical intuition as to what exactly elevated R2 values mean is not necessarily consistent with "drug interacting". Presumably, the authors are not using the helix boost component of SeqDYN here (it would be good to explicitly state this). This is not necessarily a weakness, but I think it would behove the authors to compare a few alternative models before settling on the DIRseq method, given the somewhat ad hoc modifications to SeqDYN to get DIRseq.

Actually, the factors that elevate R2 are well-established. These are local interactions and residual secondary structures (if any). The basic assumption of our method is that intra-IDP interactions that elevate R2 convert to IDP-drug interactions. This assumption was supported by our initial observation that the drug interaction propensity profiles predicted using the original SeqDYN parameters already showed good agreement with CSP profiles. We only made relatively small adjustments to the parameters to improve the agreement. Indeed we did not apply the helix boost portion of SeqDYN to DIRseq, and now state as such (p. 4, second last paragraph). We now also compare DIRseq with several alternative models, as summarized in new Table S2.

Specifically, the authors previously showed good correlation between the stickiness parameter of Tesei et al and the inferred "q" parameter for SeqDYN; as such, I am left wondering if comparable accuracy would be obtained simply by taking the stickiness parameters directly and using these to predict "drug interacting residues", at which point I'd argue we're not really predicting "drug interacting residues" as much as we're predicting "sticky" residues, using the stickiness parameters. It would, I think, be worth the authors comparing the predictive power obtained from DIRseq with the predictive power obtained by using the lambda coefficients from Tesei et al in the model, local density of aromatic residues, local hydrophobicity (note that Tesei at al have tabulated a large set of hydrophobicity scores!) and the raw SeqDYN predictions. In the absence of lots of data to compare against, this is another way to convince readers that DIRseq offers reasonable predictive power.

We now compare predictions of these various parameter sets, and report the results in Table S2. In short, among all the tested parameter sets, DIRseq has the best performance as measured by (1) strong correlations between prediction scores and CSPs and (2) high true positives and low false positives (p. 7-9).

(2) Second, the DIRseq is essentially SeqDYN with some changes to it, but those changes appear somewhat ad hoc. I recognize that there is very limited data, but the tweaking of parameters based on physical intuition feels a bit stochastic in developing a method; presumably (while not explicitly spelt out) those tweaks were chosen to give better agreement with the very limited experimental data (otherwise why make the changes?), which does raise the question of if the DIRseq implementation of SeqDYN is rather over-parameterized to the (very limited) data available now? I want to be clear, the authors should not be critiqued for attempting to develop a model despite a paucity of data, and I'm not necessarily saying this is a problem, but I think it would be really important for the authors to acknowledge to the reader the fact that with such limited data it's possible the model is over-fit to specific sequences studied previously, and generalization will be seen as more data are collected.

We have explained the rationale for the parameter tweaks, which were limited to q values for four amino-acid types, i.e., to deemphasize hydrophobic interactions and slightly enhance electrostatic interactions (p. 4-5). We now add that these tweaks were motivated by observations from MD simulations of drug interactions with a-syn (ref 13). As already noted in the response to the preceding comment, we now also present results for the original parameter values as well as for when the four q values are changed one at a time.

(3) Third, perhaps my biggest concern here is that - implicit in the author's assumptions - is that all "drugs" interact with IDPs in the same way and all drugs are "small" (motivating the change in correlation length). Prescribing a specific length scale and chemistry to all drugs seems broadly inconsistent with a world in which we presume drugs offer some degree of specificity. While it is perhaps not unexpected that aromatic-rich small molecules tend to interact with aromatic residues, the logical conclusion from this work, if one assumes DIRseq has utility, is that all IDRs bind drugs with similar chemical biases. This, at the very least, deserves some discussion.

The reviewer raises a very important point. In Discussion, we now add that it is important to further develop DIRseq to include drug-specific parameters when data for training become available (p. 12-13). To illustrate this point, we use drug size as a simple example, which can be modeled by making the b parameter dependent on drug molecule size.

(4) Fourth, the authors make some general claims in the introduction regarding the state of the art, which appear to lack sufficient data to be made. I don't necessarily disagree with the author's points, but I'm not sure the claims (as stated) can be made absent strong data to support them. For example, the authors state: "Although an IDP can be locked into a specific conformation by a drug molecule in rare cases, the prevailing scenario is that the protein remains disordered upon drug binding." But is this true? The authors should provide evidence to support this assertion, both examples in which this happens, and evidence to support the idea that it's the "prevailing view" and specific examples where these types of interactions have been biophysically characterized.

We now cite nine studies showing that IDPs remain disordered upon drug binding.

Similarly, they go on to say:"Consequently, the IDP-drug complex typically samples a vast conformational space, and the drug molecule only exhibits preferences, rather than exclusiveness, for interacting with subsets of residues." But again, where is the data to support this assertion? I don't necessarily disagree, but we need specific empirical studies to justify declarative claims like this; otherwise, we propagate lore into the scientific literature. The use of "typically" here is a strong claim, implying most IDP complexes behave in a certain way, yet how can the authors make such a claim?

Here again we add citations to support the statement.

Finally, they continue to claim:"Such drug interacting residues (DIRs), akin to binding pockets in structured proteins, are key to optimizing compounds and elucidating the mechanism of action." But again, is this a fact or a hypothesis? If the latter, it must be stated as such; if the former, we need data and evidence to support the claim.

We add citations to both compound optimization and mechanism of action.

**Reviewer #1 (Recommendations for the authors):**
(1) The authors should compare the sequences of the IDPs in the case studies with the 45 IDPs in training the SeqDYN model to make sure that they are not included in the training dataset or are highly homologous.

Please note that the data used for training SeqDYN were R2 rates, which are independent of the property being studied here, i.e., drug interacting residues. Therefore whether the IDPs studied here were in the training set for SeqDYN is immaterial.

(2) The authors manually tuned four parameters in SeqDYN to develop the model for predicting drug-interacting residues without giving strict testing or explanations. More explanations, testing of more values, and ablation testing should be given.

As responded above, we now both expand the explanation and present more test results.

(3) The authors changed the q values of L, I, and M to the value of V. What are the results if these values are not changed?

These results are shown in Table S2 (entry named SeqDYN_orig).

(4) Only one b value is chosen based on the assumption that a drug molecule interacts with 3-4 residues at a time. However, the number of interacting residues is related to the size of the drug molecule. Adjusting the b value with the size of the ligand may provide improvement. It is better to test the influence of adjusting b values. At least, this should be discussed.

Good point! We now state that b potentially can be adjusted according to ligand size (p. 12-13). In addition, we also show the effect of varying b on the prediction results (Table S2; p. 8, last paragraph).

(5) The authors add 12 Q to eliminate end effects. However, explanations on why 12 Qs are chosen should be given. How about other numbers of Q or using other residues e.g., the commonly used residues in making links, like GS/PS or A?

As we already explained, “Gln was selected because its 𝑞 value is at the middle of the 20 𝑞 values.” (p. 5, second paragraph). Also, 12 Qs are sufficient to remove any end effects; a higher number of Qs does not make any difference.

**Reviewer #2 (Recommendations for the authors):**
(1) The authors make reference to the "C-terminal IDR" in cMyc, but the region they note is found in the bHLH DNA binding domain (which falls from residue ~370-420).

We now clarify that this region is disordered on its own but form a helix-loop-loop structure upon heterodimerization with Max (p. 11, last paragraph).

(2) Given the fact that X-seq names are typically associated with sequencing-based methods, it's perhaps confusing to name this method DIRseq?

We appreciate the reviewer’s point, but by now the preprint posted in bioRxiv is in wide circulation, and the DIRseq web server has been up for several months, so changing its name would cause a great deal of confusion.

(3) I'd encourage the authors just to spell out "drug interacting residues" and retain an IDR acronym for IDRs. Acronyms rarely make writing clearer, and asking folks to constantly flip between IDR and DIR is asking a lot of an audience (in this reviewer's opinion, anyway).

The reviewer makes a good point; we now spell out “drug-interacting residues”.

(4) The assumption here is that CSPs result from direct drug:IDR interactions. However, CSPs result from a change in the residue chemical environment, which could in principle be an indirect effect (e.g., in the unbound state, residues A and B interact; in the bound state, residue A is now free, such that it experiences a CSP despite not engaging directly). While I recognize such assumptions are commonly made, it behoves the authors to explicitly make this point so the reader understands the relationship between CSPs and binding.

We did add caveats of CSP in Introduction (p. 3, second paragraph).

(5) On the figures, please label which protein is which figure, as well as provide a legend for the annotations on the figures (red line, blue bar, cyan region, etc.)

We now label protein names in Fig. 1. For annotation of display items, it is also made in the Figs. 2 and 3 captions; we now add it to the Fig. 4 caption.

(6) abstract: "These successes augur well for deciphering the sequence code for IDP-drug binding." - This is not grammatically correct, even if augur were changed to agree. Suggest rewriting.

“Augur well” means to be a good sign (for something). We use this phrase here in this meaning.

(6) page 5: "we raised the 𝑞 value of Asp to be the same as that of Glu" → suggested "increased" instead of raised.

We have made the suggested change.

(7) The authors should consider releasing the source code (it is available via the .js implementation on the server, but this is not very transferable/shareable, so I'd encourage the authors to provide a stand-alone implementation that's explicitly shareable).

We have now added a link for the user to download the source code.